# Rapid Molecular Diagnostics of Pneumonia Caused by Gram-Negative Bacteria: A Clinician’s Review

**DOI:** 10.3390/antibiotics13090805

**Published:** 2024-08-25

**Authors:** Ionela-Anca Pintea-Simon, Ligia Bancu, Anca Delia Mare, Cristina Nicoleta Ciurea, Felicia Toma, Adrian Man

**Affiliations:** 1Doctoral School of Medicine and Pharmacy, George Emil Palade University of Medicine, Pharmacy, Science and Technology of Târgu Mures, 540142 Târgu Mures, Romania; ionela.pintea-simon@umfst.ro; 2Department of Internal Medicine M3, George Emil Palade University of Medicine, Pharmacy, Science and Technology of Târgu Mures, 540142 Târgu Mures, Romania; 3Department of Microbiology, George Emil Palade University of Medicine, Pharmacy, Science and Technology of Târgu Mureș, 540142 Târgu Mures, Romania; anca.mare@umfst.ro (A.D.M.); cristina.ciurea@umfst.ro (C.N.C.); felicia.toma@umfst.ro (F.T.); adrian.man@umfst.ro (A.M.)

**Keywords:** genes, MDR, molecular biology, multiplex polymerase chain reaction, respiratory tract infection

## Abstract

With approximately half a billion events per year, lower respiratory tract infections (LRTIs) represent a major challenge for the global public health. Among LRTI cases, those caused by Gram-negative bacteria (GNB) are associated with a poorer prognostic. Standard-of-care etiologic diagnostics is lengthy and difficult to establish, with more than half of cases remaining microbiologically undocumented. Recently, syndromic molecular diagnostic panels became available, enabling simultaneous detection of tens of pathogen-related and antimicrobial-resistance genetic markers within a few hours. In this narrative review, we summarize the available data on the performance of molecular diagnostics in GNB pneumonia, highlighting the main strengths and limitations of these assays, as well as the main factors influencing their clinical utility. We searched MEDLINE and Web of Science databases for relevant English-language articles. Molecular assays have higher analytical sensitivity than cultural methods, and show good agreement with standard-of-care diagnostics regarding detection of respiratory pathogens, including GNB, and identification of frequent patterns of resistance to antibiotics. Clinical trials reported encouraging results on the usefulness of molecular assays in antibiotic stewardship. By providing early information on the presence of pathogens and their probable resistance phenotypes, these assays assist in the choice of targeted therapy, in shortening the time from sample collection to appropriate antimicrobial treatment, and in reducing unnecessary antibiotic use.

## 1. Introduction

Pneumonia is an acute respiratory infection of the lung parenchyma, involving alveoli and the distal bronchial tree. It is a major global health burden, with significant morbidity and mortality worldwide. In 2019, the total number of lower respiratory tract infections (LRTIs) was estimated at around 489 million cases, leading to 2.5 million deaths, with the highest mortality reported in the elderly (1.23 million deaths in patients of >70 years of age). LRTI was the world’s fourth leading cause of mortality and the deadliest infectious disease worldwide, surpassing global mortality due to tuberculosis and HIV infection [1].

A microbiological diagnosis of pneumonia is difficult to establish. As a result, less than 40% of hospitalized cases with community-acquired pneumonia are microbiologically documented [2]. Several factors are responsible for the low sensitivity of microbiological culture techniques in the recovery of pathogens from the lower respiratory tract: the difficulty of obtaining relevant biological specimens of adequate quality and quantity, antibiotic administration prior to sampling, the overgrowth of commensal airways microbiota, the non-detection of certain pathogens such as anaerobic or atypical bacteria, and the technical expertise of the microbiologist interpreting the cultures [3,4].

It often takes more than 48–72 h for the identification of bacteria and assessment of its antibiotic susceptibility profile through culture, and classic microbiological methods frequently fail to diagnose coinfections. Moreover, classical cultures can isolate only a small part of the etiological agents, such as cell wall-deficient bacteria, viruses or anaerobic bacteria. However, prompt therapy is essential in pneumonia patients, as delays of more than 4 h in the administration of proper antibiotic treatment is associated with worse clinical outcomes, including higher mortality rates [5]. Although the etiology of pneumonia depends greatly on epidemiological factors, as discussed below, Gram-negative bacteria (GNB) have drawn increasing attention recently, because they are associated with higher mortality [6].

A broad classification divides pneumonia cases into community-acquired pneumonia (CAP) and hospital-acquired pneumonia (HAP). HAP is defined as occurring after more than 48 h from admission to a healthcare facility, when no suspicion of disease incubation prior to admission is present [7]. Ventilator-associated pneumonia (VAP) is HAP that occurs in patients on mechanical ventilation for at least 48 h, and the invasive respiratory support is in place on the date of the event or the day before. Besides VAP, HAP includes two other particular scenarios: ventilated hospital-acquired pneumonia, which refers to severe cases of HAP that require mechanical ventilation, and non-ventilated intensive care unit-acquired (ICU) pneumonia, which is pneumonia that occurs after more than 48 h from admission in the ICU in non-ventilated patients [8]. Aspiration pneumonia occurs both in community-based and hospitalized patients, and is not regarded as a distinct entity. Pneumonitis due to aspiration of refluxed gastric acid content frequently complicates the situation, with infection caused by anaerobes [9]. Finally, health-care-associated pneumonia refers to pneumonia acquired in non-hospital care institutions [7].

Several methods have been proposed for the rapid diagnosis of pneumonia and for the evaluation of the pathogens’ resistance to antibiotics [10]. Recent advancements in the design of molecular assays enabled them to identify a wider range of targets, including viruses and bacteria, as well as frequently encountered resistance genes, all within turnaround times of a few hours.

In this narrative review, we focus on the description of commercially available molecular diagnostic tools that enable syndromic testing and antibiotic resistance profiling in the point-of-care setting, as a rapid adjunct to classical microbiological diagnostics of pneumonia due to GNB. Beside the analytical performance of the tests, we summarize the available data on their clinical utility and limitations, and also describe epidemiological data and molecular mechanisms of antimicrobial resistance relevant to the interpretation of test results. We searched MEDLINE and Web of Science for relevant articles, using as search terms: “molecular diagnostic”, “nucleic acid amplification techniques”, “polymerase chain reaction”, “PCR”, “pneumonia”, “bronchopneumonia”, “lower respiratory tract infection”, “Gram negative bacteria”, “Enterobacteriaceae”, “Enterobacterales”, “antimicrobial susceptibility”, “antimicrobial resistance”, “sensitivity”, “specificity”, “positive” or “negative” “predictive value”, as well as variations or synonyms thereof. The last search was performed on the 4 June 2024, and was limited to English-language articles. We also searched the bibliographic references of retrieved original articles and reviews for relevant papers. We selected studies according to the following criteria: reporting the performance of molecular methods in the diagnostic of pneumonia in the clinical setting (no technical notes); molecular diagnostics performed using syndromic multiplex assays applicable to respiratory infections, capable of detecting Gram-negative bacteria; commercial availability of the assay under evaluation; unequivocal description of estimates of test sensitivity and specificity (or positive/ negative percentage agreement, respectively) or reporting of data enabling the calculation of these performance measures (criterion not applied for clinical trials); and reporting of diagnostic performance of the assays for, at least, the identification of Gram-negative bacterial pathogens involved in LRTI (preferably, also the identification of antimicrobial-resistance genetic markers). For studies with overlapping datasets, the one with the largest population and the most detailed presentation of results was chosen. Studies wherein the molecular assay under evaluation was based on next-generation sequencing were excluded.

## 2. Epidemiology

Mortality in CAP is highly dependent on the treatment setting, being less than 1% in outpatient care and increasing to 4–18% in hospital wards and 50% in the ICU. Age and comorbidities further influence mortality: from 5% and 14% in patients <65 and in those >80 years of age, respectively, the risk of death is augmented by the presence of more than one comorbidity, up to 20% and 43%, for the same age groups [11].

Despite all efforts made to prevent its occurrence, HAP is the second most frequent hospital infection, accounting for more than 20% of nosocomial infections [12]. VAP is the most common form of nosocomial infection and cause of death in the ICU. Globally, the estimated mortality due to HAP and VAP is 20–30% and 20–50%, respectively [7].

The ICU-related ecosystem greatly facilitates the emergence of infections. On the one hand, patients are frequently immunosuppressed, affected by various comorbidities, have advanced age, and are submitted to invasive procedures and external-device implantation. On the other hand, the residing bacteria often manifest various degrees of antibiotic resistance and are able to fully express their virulence characters in these patients who are particularly susceptible to colonization [13].

About one third of HAP cases are caused by GNB. A large study conducted between 2012 and 2019 in the United States, comprising 253 acute-care hospitals and 17,819 patients with HAP, ventilated HAP, and VAP, found that the most common pathogens, accounting for 80% of cases, were similar across the three pneumonia types. Whereas *Staphylococcus aureus* was the most frequently isolated (40% of the cases), *Pseudomonas aeruginosa*, *Klebsiella pneumoniae* and *Escherichia coli* were the next prevalent species, each with frequencies between 9% and 20% [14]. The Survey on the Prevalence of Healthcare-associated Infections and Antimicrobial Use in Acute Care Hospitals in Spain (EPINE-2022) found *P. aeruginosa* (18.2%) and *S. aureus* (12.2%) to be the main etiological agents causing HAP, followed by *K. pneumoniae* (6.9%) and *E. coli* (6.7%). Again, data from the National Surveillance Study of Nosocomial Infection in Intensive Medicine Services in Spain (ENVIN-HELICS 2022) pointed to *P. aeruginosa* (17.5%), *S. aureus* (12.1%), and *K. pneumoniae* (10.3%), followed by *E. coli* (7.5%), *Enterobacter cloacae* (7.3%), and *Serratia marcescens* (7%) as the main pathogens involved in VAP. The relative predominance of bacteria may vary by geographic region, with *Acinetobacter baumannii* being the cause of VAP in up to 20% of cases in Eastern Europe [15].

The local ecology of the ICU influences the risk of acquiring infections with antibiotic-resistant bacteria [8]. A systematic analysis of bacterial antimicrobial resistance (AMR), covering 204 countries and 471 million individual records or isolates, estimated that AMR was responsible for 1.27 million deaths in 2019. Six pathogens (*E. coli*, followed by *S. aureus*, *K. pneumoniae*, *Streptococcus pneumoniae*, *A. baumannii*, and *P. aeruginosa*) accounted for 73.4% of deaths attributable to AMR [16]. Of the GNB, 37% are estimated to be multidrug-resistant, i.e., manifesting acquired non-susceptibility to at least one agent in three different classes. Extensive drug-resistance is defined as non-susceptibility to at least one agent in all-but-two or fewer classes of antimicrobials. VAP caused by multidrug-resistant or extensively drug-resistant microorganisms is particularly difficult to treat [8]. Patient-related risk factors for drug-resistant VAP include older age, limited mobility, immunosuppression, comorbidities such as diabetes mellitus and end-stage renal disease, malignancies, recent surgery or invasive procedures, history of prolonged hospitalization and/or long-term care facilities, current or prior ICU admission, and previously known colonization. In addition, antimicrobial therapy during the 3 months prior to admission in the ICU and recent use of broad-spectrum antibiotics markedly increase the risk (odds ratios of 13.5 and 4.1, respectively) [8,17].

A large study of the antimicrobial resistance of 28,918 consecutive isolates (one per patient) collected from patients with HAP in the USA, Western Europe and Eastern Europe (SENTRY Antimicrobial Surveillance Program), found important variations in antibiotic resistance [18]. The investigation was conducted between 2016 and 2019. GNB were responsible for HAP in more than 70% of cases from all three regions. Carbapenemase production in the *Enterobacterales* family was mainly found in *K. pneumoniae*. The phenomenon was much more frequent in Eastern than in Western Europe and the United States (23.6% vs. 1.4% and 1.7%, respectively). Regarding the type of carbapenemase, *K. pneumoniae* carbapenemase (KPC) prevalence was higher in Western Europe and the USA, whereas, in Eastern Europe, OXA-48 and metallo-beta-lactamases (MBL), mainly New Dehli metallo-beta-lactamase (NDM-1) were the more prevalent types. *E. coli* and *K. pneumoniae* (with the exception of MBL-producing strains) were susceptible to ceftazidime/avibactam in more than 90% of cases from all geographic regions. *P. aeruginosa* was found in 20.6–24.3% of patients and was highly susceptible to colistin (>99% of cases) in all three regions. Combinations of beta-lactam/beta-lactamase inhibitor, such as ceftazidime/avibactam and ceftolozane/tazobactam, followed colistin in efficacy, but with variable results: whereas in Western Europe and the USA these combinations were active on more than 93% of *P. aeruginosa* isolates, a susceptible phenotype was significantly less frequent in *P. aeruginosa* from Eastern Europe (less than 83%). *A. baumannii* represented less than 3% of isolates in the USA and Western Europe and was at least 6 times more frequent in Eastern Europe. Colistin and tobramycin were the only compounds active against more than half of isolates from all three regions. Overall susceptibility rates of *A. baumannii* were approximately 46% and 59% in Western Europe and USA, and only about 10% in Eastern Europe. A certain geographic “gradient of resistance rates” was revealed by this study, with countries in the South and East of Europe having higher resistance rates than those from the North and the West of the continent, respectively. Compared with the previous period of time, the frequencies of major GNB did not increase in either Western or Eastern Europe, with the exception of *E. coli* in the West and *K. pneumoniae* in the East. Although resistance rates for the main antimicrobial compounds decreased in Western Europe, they showed a drastic increase among key GNB in Eastern Europe, especially for *P. aeruginosa* and *K. pneumoniae* [18].

With the widespread use of antibiotics associated with the outbreak of the COVID-19 pandemic, the antimicrobial resistance rates seem to have increased [19]. Thus, among COVID patients with bacterial infections, the proportion of infections that were resistant to antibiotics was estimated at around 60.8%, while the proportion of resistant isolates was around 37.5% [20].

## 3. Main GNB Involved in Pneumonia

The lower respiratory tract (LRT) is not a sterile environment. The bacterial composition and diversity in LRT samples can be assessed by studying the lung microbiome via 16S rRNA gene amplicon analysis. A small group of bacteria, composed of *Streptococcus* spp., *Prevotella* spp., *Veillonella* spp., *Haemophilus* spp., and *Fusobacterium* spp., found in most healthy individuals, seems to be essential for lung homeostasis and was proposed as a core pulmonary microbiota [21,22]. Centralization of 16S rRNA sequencing data from 17 published studies including 2177 respiratory samples collected from 1029 critically ill patients (21.7% with ARDS and 26.3% with HAP) and 327 healthy controls suggested that microbial signatures for HAP and ARDS are characterized by a relative depletion of the members of this core group, concomitant with an enrichment of potentially pathogenic bacteria such as *Staphylococcus* and *Pseudomonas* [21]. Fenn et al. described the lung microbiome in patients suspected of VAP. Despite the obvious difference in detection sensitivity between deep sequencing and microbiological culture (over 80 genera detected vs. 8 genera isolated, respectively), the top-three most frequently identified genera by the two techniques in bronchoalveolar lavage (BAL) samples were identical: *Staphylococcus*, *Pseudomonas* and *Haemophilus*. In patients with positive culture, an increased dysbiosis of the lower respiratory tract microbiome was observed, together with an increased prevalence of pathogenic bacteria, increased genus dominance and decreased diversity, compared with patients with negative cultures [23].

A certain difference in the distribution of causative bacteria in CAP, HAP and VAP was classically documented, which served as a guidance for the choice of empiric therapy, according to the epidemiological context of a patient. Thus, in CAP, GNB are not described as common bacterial causes; *S. pneumoniae*, *Mycoplasma pneumoniae*, *Chlamydia pneumoniae*, *H. influenzae* and *S. aureus* are involved instead. In HAP, GNB such as *P. aeruginosa*, *Klebsiella* spp., *Acinetobacter* spp., and *Enterobacter* spp. are common etiologies, whereas in VAP these pathogens differ in their frequency of involvement: *P. aeruginosa*, *S. aureus*, *Acinetobacter* spp., *Klebsiella* spp., and *Enterobacter* spp. [5].

*A. baumannii* is a ubiquitous, non-fermentative, Gram-negative coccobacillus that produces infections in critically ill patients. Severe respiratory infections caused by this agent can lead to bloodborne dissemination and systemic infections. Multiple resistance mechanisms were described in *A. baumannii*: the production of carbapenemases, chromosomally encoded AmpC cephalosporinases, aminoglycoside-modifying enzymes, and extended spectrum beta-lactamases, as well as the ability to diminish the intrabacterial antibiotic concentrations through porin downregulation and efflux overexpression [13,24].

*K. pneumoniae* is a Gram-negative short bacillus that usually inhabits the human gut, but occasionally colonizes the upper respiratory tract and other anatomical sites. Patients with *K. pneumoniae* colonization on admission in the ICU show high probability of developing severe infection with the same microorganism. This is why the screening of patients for this bacterium, as for other potentially resistant species such as MRSA or VRE, is strongly recommended upon admission to the ICU. Acute infections produced by *K. pneumoniae* are enabled by the wide spectrum of virulence factors it possesses. For example, the gene loci for salmochelin and aerobactin synthesis contribute to siderophore production, allowing the bacterium to live in iron-deficient conditions. Enterobactin has the highest iron affinity among siderophores, and is expressed in both classical and hypervirulent strains [25]. *K. pneumoniae* bacteria present polysaccharide capsules that protect them from the host’s defense mechanisms. Increased capsule production regulated by the *rmpA* gene leads to the expression of a hypermucoviscous phenotype with enhanced virulence against the human host. In addition, these bacteria can make use of fimbriae and biofilm-related adhesins in order to colonize various anatomic niches. *K. pneumoniae* manifests a wide variety of antimicrobial-resistance patterns. The main phenomenon is carbapenem-resistance, due to the production of several beta-lactamases and/or alteration of membrane permeability [26]. Simultaneous expression of carbapenem-resistance genes and virulence-encoding genes can lead to the occurrence of particularly difficult-to-combat strains [27].

*P. aeruginosa* is a non-fermenting, Gram-negative opportunistic pathogen that is a leading cause of hospital-related infections. Its high virulence is facilitated by type IV pili and flagella, and the outer membrane structures that produce lipopolysaccharide or alginate as a biofilm-related compound. Biofilm formation enables it to survive in the environment, even in difficult growth conditions. Biofilm formation and the expression of virulence traits are regulated by the quorum-sensing phenomenon [28]. Several antibiotic resistance mechanisms have been described in *P. aeruginosa*, including production of MBL or KPC, leading to carbapeneme non-susceptibility, as well as membrane permeability alterations via the overexpression of efflux pumps or depression of the outer membrane protein (OMP) [29].

*E. coli*, similarly to *K. pneumoniae*, resides in the gastrointestinal tract, but can migrate to extra-intestinal sites, a phenomenon correlated with the production of virulence factors: iron acquisition factors, adhesins, invasins, lipopolysaccharides, toxins, and the polysaccharide capsule. Being an opportunistic pathogen, it most commonly affects immunosuppressed and debilitated patients. Among resistance mechanisms in *E. coli*, carriage of extended-spectrum beta-lactamase (ESBL) encoding genes (*bla* genes) is the most common. The *bla*_CTX-M_ genes, along with *bla*_NDM_ and *bla*_KPC_, are the most clinically relevant resistance markers in this microbe [30].

Because of the threat that they pose to global public health, multidrug resistant *A. baumannii*, *P. aeruginosa* and *Enterobacteriaceae* (including *E. coli* and *K. pneumoniae*) are included in the Priority 1 (critical) category of the WHO list of “priority pathogens”, which are increasingly difficult to treat [31].

### Phenotypes and Genotypes of Antimicrobial Resistance

GNB account for 10 out of the 18 antibiotic-resistant threats identified by the CDC [32]. The term “difficult-to-treat resistance” refers to GNB manifesting resistance to all fluoroquinolones and all β-lactam categories, including carbapenems. Seriously challenging phenotypes found in clinical practice now include carbapenem-resistant *P. aeruginosa*, carbapenem-producing *Enterobacterales* (CPE, in particular producers of KPC, MBL, and oxacillinase [OXA]-type carbapenemases), carbapenem-resistant *A. baumannii* and third-generation cephalosporin-resistant GNB [17].

Antibiotic-resistance mechanisms have been classified into four categories: drug inactivation, alteration of drug target, limitation of drug uptake, and increased levels of drug efflux [33]. GNB are able to acquire all of these mechanisms of resistance, some of which are simultaneously expressed in the same strain.

The production of beta-lactam hydrolyzing enzymes is a major resistance mechanism in GNB. Beta-lactamases are clinically relevant because they drive resistance against several important classes of antimicrobials used in medical practice, such as penicillins, cephalosporins and carbapenems. Among beta-lactamases, three groups are of particular clinical importance: ESBL, AmpC beta-lactamases, and carbapenemases [34].

ESBLs are active against penicillins, the first three generations of cephalosporins and aztreonam, but do not hydrolyze carbapenems and cephamycins (cefoxitin). ESBLs are inhibited in vitro by beta-lactamase inhibitors such as clavulanate. These enzymes are generally encoded by genes harbored in plasmids also carrying other resistance determinants, which can be passed among various species of *Enterobacteriaceae* [34].

Carbapenemases belong to three classes (A, B, and D), according to the Ambler classification. Carbapenemase class A, such as KPC, can hydrolyze almost all beta-lactams. The most common subtypes of *bla*_KPC_ are *bla*_KPC-2_ and *bla*_KPC-3_ in the USA and *bla*_KPC-2_ in Europe and Asia. Class B carbapenemase enzymes are also named MBL, since they present one or two zinc ions as active centers. This class includes New Delhi MBL (NDM-1), Verona integron-encoded MBL (VIM-1) and imipenemase (IMP). MBLs can hydrolyze nearly all β-lactam antibiotics, except for monobactams (such as aztreonam). Class D enzymes, which include oxacillinase 48 and 181 (OXA-48 and OXA-181) have a weak carbapenemase activity [27,35].

AmpC beta-lactamases hydrolyze penicillins, cephalosporins of the first to the third generation, and, frequently, aztreonam. Unlike ESBLs, they are not inhibited by beta-lactamase inhibitors in vitro, and are able to degrade cephamycins. These enzymes are encoded on the bacterial chromosome or in plasmids. Chromosomal expression of AmpC in *Enterobacteriaceae* (seen especially in *Enterobacter* spp., but also in *S. marcescens*, *Citrobacter freundii*, *Providencia* spp. and *Morganella morganii*) is usually induced by the use of cephalosporins. The consequence is that, although the AmpC-carrying strain may seem susceptible to cephalosporins in vitro, the clinical use of this class of antimicrobials may be associated with treatment failure. Plasmid-mediated AmpC is frequently seen in *E. coli* and *Klebsiella*. These plasmids often encode for other non-related resistance genes [34].

Besides antibiotic degradation, bacteria can gain resistance by decreasing antimicrobial drug intracellular concentration. Two mechanisms contribute to this effect. On the one hand, activation of the efflux pump system can lead to the rapid pumping of antibiotic out of the bacterial cell. In *K. pneumoniae*, upregulation of the transcription factor ramA enhances the expression of *acrAB* and *tolC* genes, increasing the production of efflux pump proteins AcrAB-TolC, eventually leading to multidrug resistance. On the other hand, decreasing expression of outer membrane porins limits the amount of antibiotic entrance in the bacteria. OmpC, OmpD, OmpE, OmpF, and PhoE are common OMPs encountered in *Enterobacter* strains of clinical importance. In *K. pneumoniae*, OmpK36 belonging to the OmpC family and OmpK35 belonging to the OmpF family, are usually found in carbapeneme-resistant strains. For example, high levels of resistance to ertapenem were seen in *K. pneumoniae* with single point mutations or presence of insertion sequences in *ompK35* and *ompK36* genes, leading to the absence of expression of the corresponding porins [27]. The spread of various carbapenemase types differs by geographic region. For example, KPC predominates in the USA and Western Europe (84.7% and 68.3% of isolates, respectively), whereas the most common carbapenemases found in the Eastern Europe are OXA-48 and MBLs, mainly NDM-1 [18].

Occasional plasmid fusion or recombination events leading to the integration of carbapenem resistance and hypervirulence phenotypes in the same bacterial cell confer enhanced pathogenic potential to the affected strains [27].

With the development of targeted antimicrobial therapies, knowledge of the beta-lactamase involved in the resistance phenotype of a particular strain is increasingly important in clinical practice. Recently, three novel combinations of beta-lactam/beta-lactam inhibitor have been developed for the replacement of colistin and polymyxin B in the treatment of CRE and multidrug-resistant *P. aeruginosa*: imipenem/relebactam, ceftazidime/avibactam, and meropenem/vaborbactam. Relebactam was developed to enhance the activity of imipenem against carbapenem-resistant *P. aeruginosa* and CRE expressing KPC. Vaborbactam introduction sought the direct inhibition of KPC, whereas avibactam inhibits both ESBLs, KPC and OXA-48 [36]. It is worth noting that the combination meropenem/vaborbactam is not active against bacteria that produce carbapenem-hydrolyzing oxacillinases and MBLs, as demonstrated by the different exposure responses in isolates expressing OXA-48 vs. those expressing KPC [36]. Also, phenotypic breakpoints were found to be less reliable in the clinical use of meropenem and meropenem/vaborbactam against OXA-48 producers [37]. Ceftolozane, a modified cephalosporin, associated with the beta-lactamase inhibitor tazobactam, is effective against ESBL-producing Enterobacterales, against GNB with overexpression of AmpC, porin loss, or drug efflux pumps, and is the most potent beta-lactam against *P. aeruginosa* that does not produce MBLs or serine-carbapenemases [37].

## 4. Molecular Diagnostic Assays

Conventional microbiological diagnostic techniques for pneumonia rely on the cultivation of the microorganisms present in the clinical sample, using specific culture media. Usually, bacterial growth takes 24 to 48 h. Then, aspects such as culture purity, microbial quantity and morphology are evaluated. In respiratory infections, microbial count is linked to the etiology of infection, since multiple bacterial types are normally present in the lower respiratory tract. Usually, a threshold count is used, over which bacterial growth is considered significantly associated with the suspicion of infection. Then, antimicrobial susceptibility testing (AST) is performed on significantly growing microorganisms, which adds another 24 h. Importantly, AST results provide the minimum inhibitory concentration values needed in the planning of the targeted antimicrobial therapy. In total, 72 h are often needed for a complete microbiological diagnostic. The presence of fastidious microorganisms, polymicrobial growth and extended resistance patterns of the isolated microbes frequently prolong the time needed for diagnostics [13]. Because of long turnaround times, the requirement for viable pathogens present in the clinical sample, an inability to identify viruses, and difficulties in identifying atypical bacteria or microorganisms that need special growing conditions, conventional microbiological diagnostics is considered to have a rather low diagnostic sensitivity in lower respiratory infections [38].

Molecular diagnostic techniques, especially those that employ nucleic acid amplification, may have certain advantages over classical methods. They provide high sensitivity and specificity in the identification of respiratory pathogens, are able to simultaneously detect and quantify multiple pathogens from the same sample, and have the potential to assess antimicrobial resistance patterns [6]. Amplification of the target nucleic acid, performed either by PCR or isothermal techniques, relies on enzymatic reactions which do not require that the pathogens in the sample be viable, meaning that the onset of antibiotic treatment prior to sample collection is less likely to affect the results. Viruses and non-cultivatable pathogens can be easily identified by their nucleic acid signature, such that PCR-based techniques, preceded or not by reverse transcription, became the gold-standard for the diagnosis of infections provoked by these agents. Assays can be relatively rapidly updated to cope with the emergence of genetic mutations in target pathogens. Amplification products can be identified by a variety of methods, including the use of hydrolysable fluorescent probes (e.g., Fast Track Diagnostics Respiratory Pathogens 21 Assay, Seegene Allplex Respiratory Panel Assays, Qiagen QIAstat-Dx Respiratory SARS-CoV-2 Panel), melting curve analysis (e.g., Biomérieux Biofire Filmarray Pneumonia Panel), and solid-phase hybridization (e.g., Diasorin Verigene Respiratory Pathogens Flex Test, Curetis Unyvero Hospitalized Pneumonia). Depending on the conditions, target amplification and detection may be completed within minutes [39,40].

The BioFire FilmArray Pneumonia Panel plus, Biomérieux, is a closed, pouch-based, multiplexed nucleic acid test designed to detect and identify several respiratory bacterial and viral pathogens, as well as certain genetic markers of antimicrobial resistance. Intended to be used on dedicated commercial platforms, the disposable device contains all the needed reagents for the automatic sample lysis, nucleic acid isolation, amplification and detection. Amplification is performed via nested PCR in two steps, comprising an initial highly multiplexed PCR reaction, followed by singleplex reactions for the individual targets. Identification is performed by melting curve analysis, whereas data from real-time amplification kinetics are used for the relative quantification of bacterial targets in relationship to an internal standard, which enables the approximation of their respective genomic copy numbers per milliliter of sample. Appropriate specimen types include BAL-like specimens (BAL and mini-BAL) and sputum-like specimens (induced and expectorated sputum, and endotracheal aspirate, ETA). The panel allows the detection of 15 typical bacteria, 3 atypical bacteria, 7 resistance genes (see Table 1), and 8 viruses (adenovirus, coronavirus (229E, OC43, HKU1, NL63, Middle-East Respiratory Syndrome-associated coronavirus, but not the severe acute respiratory syndrome coronavirus 2, SARS-CoV-2), human metapneumovirus, human rhinovirus/enterovirus, influenza viruses A and B, parainfluenza virus and respiratory syncytial virus). The earlier version of the assay (i.e., BioFire FilmArray Pneumonia Panel) did not report the presence of MERS-CoV.

The 15 typical bacteria are reported as DNA copies/mL by approximation to the nearest integer log (10^4^,10^5^,10^6^ or >10^7^). Bacteria with calculated DNA copies/mL of less than 3.5 log_10_ are reported as “not detected”. Results of antimicrobial resistance gene detection are reported conditionally on the detection of a plausible microorganism able to carry the respective markers. Thus, the presence of mecA/C and MREJ is only reported if *S. aureus* is detected (a combination indicative of the presence of methicillin-resistant *S. aureus*, MRSA). The presence of OXA-48-like is reported if at least one of the following bacteria is found: *Enterobacter cloacae* complex, *E. coli*, *Klebsiella aerogenes*, *Klebsiella oxytoca*, *Klebsiella pneumoniae* group, *Proteus* spp., and *S. marcescens*. The rest of the six antimicrobial resistance genes (CTX-M, IMP, KPC, NDM, VIM) are reported for *Acinetobacter calcoaceticus-baumannii* complex, *E. cloacae* complex, *E. coli*, *K. aerogenes*, *K. oxytoca*, *K. pneumoniae*, *P. aeruginosa*, *Proteus* spp., and *S. marcescens* [41]. Of note, the detected resistance marker and the detected microorganism cannot be linked together.

The Unyvero Hospitalized Pneumonia (HPN) Application, Curetis GmbH, is another commercially available multiplex diagnostic system for use on lower respiratory-tract specimens. The test cartridge is processed on two different instruments of the Unyvero platform, to complete the sample lysis and the amplification/ detection of the target nucleic acid sequences. Like the FilmArray, the HPN uses bead beating to efficiently homogenize the sample and break the bacterial walls. Amplification is performed in eight multiplex PCR reactions of four to six targets each, whereas detection is made following hybridization of amplicons on a membrane array. HPN detects 17 typical bacteria, 3 atypical bacteria, 1 fungus and 19 genetic resistance markers (Table 1). It does not include viral targets. Results of microbial target detection are presented semi-quantitatively as “not detected” or “detected” (from + to +++). A number representing the relative intensity of the hybridization signal is also displayed. This number is related to the detection process and does not reflect the microbial DNA copy number in the sample. Among the antibiotic resistance markers detected, 15 refer to transferable genetic elements. The detection of these markers cannot be linked with a specific microorganism, but the results report indicates groups of bacteria that may carry these genes, according to the literature. Four chromosomal (non-transferable) resistance loci are examined by the assay, namely codons 83 and 87 of gyrase subunit A (*gyr*A83 and *gyr*A87) in *E. coli* and *P. aeruginosa*. More specifically, the test detects the wild-type sequence of the gene at the specified loci, which reflects sensitivity to gyrase inhibitors/fluoroquinolones. Absence of detection of one of the targets in this case means antimicrobial resistance. If one of the *gyr* markers is detected, but *E. coli* or *P. aeruginosa* are not detected or are below the reportable threshold, the specific *gyr* result is marked as “cannot be interpreted” [42].

Like the FilmArray and the Unyvero HPN, QIAstat-Dx Respiratory SARS-CoV-2 Panel, Qiagen, is another “sample-to-answer” cartridge-based multiplex assay for respiratory pathogens. It is designed to perform all steps, from nucleic acid purification to detection, on one instrument. Detection is achieved via multiplex real-time PCR in about 75 min and, unlike the other two platforms, individual amplification curves for all detected targets are provided, aiding in their interpretation and quantification. The assay covers 19 viral targets and 4 bacteria (*M. pneumoniae*, *L. pneumophilia*, *B. pertussis*, *C. pneumoniae*). It does not provide information on antimicrobial resistance markers [43]. In addition to the earlier version (V1), V2 includes the detection of C. pneumoniae.

The Fast Track Diagnostics (FTD) Respiratory pathogens 21 panel, Siemens Healthcare Diagnostics Inc., differs from the first three tests discussed, in that it is exclusively a laboratory-centered assay. It is able to detect 20 viruses and 1 bacterium (*M. pneumoniae*) in five parallel reactions [44]. Sample preparation, nucleic acid extraction, and amplification/detection are performed separately by laboratory staff. Unlike the all-in-one cartridge or pouch formulations whose use needs very little expertise and hands-on time, the use of the FTD respiratory panel involves highly trained personnel and considerably longer hands-on time. On the other hand, amplification and detection via multiplex real-time PCR proposed by this assay can be performed on a variety of instruments and is not dependent on one producer’s highly specialized closed system. However, the assay cannot be performed as a point-of-care test or in the Emergency Department.

Due to the low coverage of GNB by the QIAstat-Dx and FTD respiratory panels, these tests will not be discussed in further detail in this review.

## 5. Diagnostic Performance of Rapid Molecular Tests in GNB Pneumonia

### 5.1. Identification of Respiratory Bacterial Pathogens

Multiplex PCR-based assays for LRT infections show good diagnostic performance and concordance with bacterial culture results, regarding detection of respiratory pathogens (Table 2 and Table 3).

Curetis was the first to launch a multiplexed pneumonia panel in 2012, with the Unyvero Lower Respiratory Tract panel [5]. An early version of the test (the P55 Pneumonia cartridge) showed rather low sensitivity and specificity (56.9% and 58.5%, respectively) when compared with standard methods on 74 BAL specimens from patients admitted to the ICU [45]. Another evaluation of the same product on 95 respiratory samples, compared with microbial culture and subsequently verified by 16S sequencing, reported better estimates of sensitivity and specificity (88.8% and 94.9%, respectively) [46]. The difference, however, may be attributable to the different storage conditions of the samples: whereas the latter study employed fresh samples (less than 48-h old), the former used samples frozen at −80 °C, stored for 8 to 41 months. In this case, long-term storage may have affected the quality of the samples and. implicitly, the reliability of results. However, even on fresh samples, the system exhibited a relatively high failure rate, affecting 10.5% of the processed samples [46]. The FDA-cleared version of the Unyvero Lower Respiratory Tract panel reduced the threshold for reporting positive detection results to the equivalent of 10^4^ CFU/mL, greatly improving the diagnostic performance of the assay. When evaluated on 175 BAL samples, the test showed a positive percentage agreement (PPA) of 96.5% and a negative percentage agreement (NPA) of 99.6% with the culture results [47]. As expected, the Unyvero panel had more positive results than the culture. During discrepant analysis by sequencing, more than 70% of the results initially considered false positives were shown to actually present the DNA of the detected microorganism.

In a retrospective study on 659 samples (two-thirds of samples BAL, one-third ETA) from patients suspected of pneumonia, the Unyvero Lower Respiratory Tract panel had an overall sensitivity of 85.7% and an overall specificity of 98.4% [48], compared with the culture. The sensitivity of the assay varied for different microorganisms, in some cases being below 80%, for example: 77.8% for *S. pneumoniae* and *S. marcescens*, 76.2% for *K. oxytoca*, 74.5% for *K. pneumoniae*, and 71.4% for *H. influenzae*. Regarding antimicrobial resistance, the Unyvero LRT panel predicted no change in antibiotic treatment in 12.4%. However, according to the panel results, antibiotic de-escalation would have been proper for 65.9% of patients, of which 64% had unnecessary coverage for P. aeruginosa and 69% had unnecessary MRSA coverage [48].

A multicenter study with 1016 prospectively collected and 392 archived specimens from 11 clinical trial sites in the United States, evaluated the Unyvero Lower Respiratory Tract BAL Application (Curetis GmbH) in comparison with the standard-of-care microbiological diagnostics [49]. In the case of discrepant results, samples were analyzed by PCR followed by bidirectional Sanger sequencing. For typical bacteria, the PPA and NPA of the molecular assay with routine culture were 93.4% and 98.3%, respectively. Individual PPA values were above 90% for all on-panel analytes, with the exception of *K. pneumoniae* (89.1%), *E. cloacae* complex (77.8%) and *K. variicola* (50%). Individual NPA values ranged between 95.2% and 99.8% for microorganisms in prospective samples. A high rate of additional detections (considered false positives) was made by the Unyvero application compared with the culture, in prospectively collected specimens (21.7%). In most of these cases (84.9%), the presence of the identified microorganism DNA was subsequently confirmed by PCR/sequencing. Many of the additional confirmed detections were *Acinetobacter* spp., *S. aureus* or *P. aeruginosa*, considered clinically relevant pathogens. In 2.2% of prospective samples, the Unyvero failed to detect microorganisms retrieved by culture (false negative results).

In a study on 83 lower respiratory-tract samples from critically ill COVID-19 patients, the Unyvero HPN Application had 95.1% (95% CI 86.5–98.3%) sensitivity and 98.3% (95% CI 97.5–98.9%) specificity for the detection of bacterial pathogens, compared with the culture [4]. The panel failed to detect bacteria in 3.6% of tested samples and detected additional bacterial species in 25.3% of samples.

A multicenter prospective observational study organized in 2018 evaluated the BioFire FilmArray Pneumonia Panel in comparison with routine conventional methods for the microbiological diagnostics of pneumonia [50]. A total of 515 respiratory samples from consecutive patients with suspected pneumonia were tested. The index test showed a PPA value of 94.4% (95% CI 91.7–96.5%) and an NPA of 96.0% (95% CI 95.5–96.4%) when compared with the culture. The FilmArray detected typical bacterial targets in 68.5% of specimens. While detecting most bacterial pathogens isolated by culture (94.4%), the test also detected additional bacteria in 37.7% of samples. Since the bacterial culture was considered the gold standard, the additional findings by FilmArray were deemed false positives, leading to a rather low positive predictive value (PPV) of 56.0% (95% CI 52.1–59.8%). However, negative predictive values (NPV) were very high overall: 99.7% (95% CI 99.5–99.8%). The best agreement between FilmArray and the culture was observed for *P. aeruginosa*, *K. pneumoniae* group, *K. aerogenes* and *S. aureus*. Among false negative results given by the FilmArray, most pathogens belonged to species not covered by the panel (46 isolates, including 10 *Citrobacter* spp., 9 *S. maltophilia*, 9 *M. morganii*, and 9 *Hafnia alvei*). Twenty-two bacteria isolated by culture, mostly *Enterobacteriaceae* (n = 15, 68%) were not detected by the panel, although they were included among its targets.

The performance of the BioFire FilmArray Pneumonia Plus panel, compared with standard-of-care testing, was assessed in a large multicenter study involving 52 laboratories from 14 countries [51]. A total of 2476 split samples consisting of fresh or frozen aliquots of BAL-like and sputum-like specimens were tested. In total, 3893 pathogens were detected by at least one method, of which FilmArray detected 96.15%, while standard-of-care diagnostics identified 51.25%. The largest discordances were observed in the case of fastidious bacteria, which may lose viability in the course of standard testing or may be concealed by normal flora: *S. pyogenes* (in 77.27% of discrepant samples), *S. agalactiae* (78.85%), *M. catarrhalis* (61.7%), *H. influenzae* (53.97%), and *S. pneumoniae* (48.07%). The overall concordance between methods was 47.39%. Mean PPA for the common bacterial pathogens was 92.90% (95% CI 91.61–94.04%) and mean NPA was 96.10% (95% CI 95.89–96.30%) for all specimens combined. Again, due to the higher detection sensitivity of FilmArray compared to the reference standard-of-care testing, PPVs were modest, whereas NPVs were, on average, 99.63% (95% CI 99.56–99.68%) for bacterial targets, all specimens combined. The FilmArray performed similarly on BAL-like and on sputum-like specimens. In 45.1% of false negative results by FilmArray, standard diagnostics reported low levels of bacteria (e.g., 1 colony, <10^3^ CFU/mL), which fall below the detection threshold of the multiplex PCR panel. Standard testing identified 649 potential pathogens not detected by FilmArray. Of these, 230 were GNB, including 70 isolates of *S. maltophilia*, 31 *Citrobacter* spp., and 28 *Acinetobacter* spp. Yeasts were found in 283 specimens, including 123 isolates of *Candida* spp.

The agreement between multiplex PCR and microbial culture varies with the targeted species, suggesting different analytical sensitivity of the primers and probes employed by the molecular assay. For example, Gong et al. found a PPA of 100% for *H. influenzae*, *P. aeruginosa*, *S. marcescens*, and *S. aureus*, and 73% for *K. pneumoniae*, while comparing the BioFire FilmArray Pneumonia panel with the culture in LRTI [52].

Generally, in published studies, the percentage of false negative results reported by FilmArray due to the presence of bacterial respiratory pathogens not included in the molecular panel is below 10%. For example, in a summary of 15 pooled studies including 4596 respiratory samples from over 4200 patients, Dessajan and Timsit found that 9% of the pathogens involved were absent from the FilmArray Pneumonia Plus Panel [3]. For the same studies, the average sensitivity for bacterial identification by multiplex PCR compared with standard culture was 92%, with a specificity of 97% [3]. These figures seem to be pretty good estimates of the test performance, and are similar to the overall sensitivity and specificity of 94% (95% CI 91–95) and 98% (95% CI 97–98), respectively, reported in a recent meta-analysis of FilmArray results in approximately 9000 respiratory specimens [53].

A side-by-side comparison of the two main PCR-based multiplex panels, BioFire FilmArray Pneumonia Panel (bioMérieux) and Unyvero Hospitalized Pneumonia Panel (Curetis), was performed in a multicenter study involving 652 LRT samples from patients with suspected HAP/VAP in 15 intensive care units from different UK hospitals [54]. Because of the limitations of culture methods alone, the authors used Bayesian latent class analysis, incorporating information from both PCR panels and routine microbiology, to assess sensitivity and specificity of the tests. Molecular assays identified pathogens in considerably more samples compared with routine microbiology: 60.4% and 74.2% for Unyvero and FilmArray, respectively, vs. 44.2% by routine microbiology. Variations according to microbial species were observed in this regard. *E. coli* and *Klebsiella* spp. were detected relatively more frequently by PCR, whereas *S. aureus* and *P. aeruginosa* were found less frequently. The difficulty of bacterial wall lysis in certain species, required for the amplification of the target DNA by molecular methods, may be, at least in part, responsible for these differences. For common HAP/VAP pathogens, Unyvero had sensitivity of 50.0–100.0%, and specificity of 89.5–99.0%, whereas FilmArray had sensitivity of 91.7–100.0% and specificity of 87.5–99.5%. By the same analysis, routine microbiology had low sensitivity, ranging from 27.0% to 69.4%. While sensitivity of both panels was above 90% for most bacterial targets, sensitivity of FilmArray seemed to be higher than that of Unyvero in some targets, e.g., *E. coli* (98.9% vs. 89.6%), *K. aerogenes* (89.8% vs. 48.4%), *K. pneumoniae* (98.1% vs. 88.9%), and *S. marcescens* (97.1% vs. 90.8%). It is possible that the nested-PCR design which the FilmArray relies on, offers it a higher sensitivity than Unyvero’s, as seen before in the comparison of other respiratory panels [55]. Neither of the two PCR-based tests’ performance was influenced by the pre-analytical factors investigated: the timing of sample collection related to antibiotic administration, the fresh or frozen status of the sample, or the time from sample collection to testing (up to 72 h) [54].

### 5.2. Identification of Bacterial Resistance to Antimicrobials

Although the ability of commercial multiplex PCR panels to identify bacterial pathogens in respiratory samples has been widely studied, data regarding their diagnostic performance in the detection of AMR genes (Table 2 and Table 3) are more scarce, either because comparison of panel results with antimicrobial susceptibility testing by conventional methods was not reported [51], or the prevalence of drug-resistant bacteria in the studied patients was low. For example, in a study conducted in Switzerland, in 1078 BAL samples, 393 bacterial targets were detected. Among these, only 11 AMR determinants were identified by FilmArray (5 CTX-M and 6 mecA/C and MREJ). Conventional AST confirmed two ESBL- producing *Enterobacterales* and three MRSA, and did not detect additional resistant bacteria [56].

In 515 respiratory specimens from a multicentric study, the FilmArray Pneumonia Panel detected 42 resistance markers [50]. Of these, 24 (50%) were confirmed by phenotypic testing: 17 ESBL-producing Enterobacteriaceae and 7 MRSA. In addition, the molecular assay reported the presence of CTX-M and VIM carbapenemase in six and two samples, respectively, none of which presented Gram-negative bacteria in culture. Conversely, two instances of ESBL production were not reported by the assay, one of *C. freundii* and one of *M. morganii*, since these pathogens are not covered by the panel.

Gadsby et al. reported poor prediction of phenotypic resistance to amoxicillin, third-generation cephalosporin and macrolide/ lincosamide by the Unyvero P55, with 8/25 (32%), 0/8 (0%) and 0/8 (0%) resistant isolates detected respectively [45]. Moreover, 80% of fluoroquinolone resistance cases indicated by the molecular assay were false positive (four out of five).

In a study of 93 suspected VAP episodes in 83 patients, resistance mechanisms identified by the Unyvero platforms (P55 and HPN were used) were concordant with conventional AST in 62 cases, but differed in 31 episodes (33%) [57]. Unyvero’s failure to detect resistance was responsible for 71% of discordances. Resistance identification failed more frequently when *P. aeruginosa* was present (24 of the 31 episodes): there were seven cases of false resistance to fluoroquinolones reported by the Unyvero, whereas resistance to carbapenems and third-generation cephalosporins was not detected in five and twelve cases, respectively.

In the multicentric study reported by Klein et al. (1016 prospective LRT specimens), resistance markers detected by Unyvero LRT were confirmed by PCR/sequencing in 95.7%, 100%, 95% and 72.5% of the corresponding specimens, for the presence of CTX-M, carbapenemase genes, TEM and mecA, respectively [49]. However, the absence of a certain gene for AMR does not always translate into susceptibility of the corresponding antibiotic, as other resistance mechanisms may be present in the microorganism. In the case of several microorganisms present in a sample, the authors only compared Unyvero results of resistance genes with data of phenotypic resistance if results of conventional AST were available for all pathogens isolated from the sample. For this limited subset of 12 CTX-M positive samples, Unyvero’s phenotypic PPV was 100%. Among the *Acinetobacter* isolates, a phenotypic PPV of 88.9% was observed (8/9 isolates): Unyvero reported three cases of OXA-23 and five cases of OXA-24, all of which proved carbapenem-resistant; in the case of a discordant result, the molecular panel reported the presence of OXA-24, but the isolate was carbapenem-susceptible.

In a study of 95 clinical samples from patients with ventilated HAP or VAP, the Unyvero HPN correctly identified five of eight (63%) extended-spectrum beta-lactamases (CTX-M gene) and four of four (100%) carbapenemases genes (three NDM, one OXA-48) [58].

Enne at al. compared the FilmArray Pneumonia Panel and Unyvero Hospitalized Pneumonia Panel for their ability to identify the presence of genes encoding ESBLs and carbapenemases in 652 clinical samples [54]. Among 17 *Enterobacterales* with ESBL phenotype, the CTX-M gene was detected by Unyvero in 12 samples, and by FilmArray in all 17 samples. However, false positive ESBL prediction was made by Unyvero in 2 samples and by FilmArray in 15 samples. Regarding carbapenemase production, the culture confirmed a resistant phenotype in seven of the eleven detections by Unyvero, and in two of the three detections by FilmArray. There were eight carbapenem-resistant isolates identified by the culture. Unyvero detected carbapenemase genes in all of them, whereas FilmArray only identified two.

Interpretation of antimicrobial resistance prediction by PCR-base methods is complex and should be made with caution, especially in settings with low prevalence of resistance. Being dependent on prevalence, the NPV of the molecular assays is high, but their PPV is low. In other words, in this case, detection failure (false negative results) would have little impact, with few cases missed. However, the probability of overdiagnosis (false positive) is much higher. An overview of 15 studies pointed out that, in the case of CTX-M being present in 4% of isolates, the probability of false positive detection by genetic testing would be 27.6%, while for a prevalence of OXA-48-like equal to 0.2%, overdiagnosis would be higher than 35% [3].

### 5.3. Discordances between the Multiplex-PCR Panels and Conventional Culture in the Detection of GNB in Respiratory Samples

The clinical significance of the high detection rates by PCR is still being debated.

As already stated, differences between conventional culture results and molecular panels may arise from the difficult identification of low-concentration pathogens in polymicrobial samples, or from poor recovery of fastidious microorganisms [5]. Antibiotic exposure, as well as various metabolic stressors, may cause bacteria to enter a “dormant” state, also designated a “viable but non-culturable state” (VBNC). Even though bacterial cells in this condition are still alive, demonstrated by their ability to synthetize RNA and sustain metabolic activities, they become temporarily unable to produce colonies on routine culture media, and thus remain undetectable by classical microbiological methods. However, they retain much of their virulence and many of their antimicrobial resistance traits. Upon resuscitation, these microorganisms are able to induce an infectious process in their host. Common human pathogens that may enter a VBNC state include *E. aerogenes/cloacae*, *E. coli*, *K. pneumoniae*, *P. aeruginosa*, and *S. marcescens*. VBNC bacteria can be detected by molecular methods, by identification of their specific sequences of DNA or RNA (gene expression) [59].

In a comparison between a multiplex PCR panel (Unyvero Hospitalized Pneumonia) and bacterial culture, it was observed that the percentage of results initially considered false positives for the molecular test (29%) significantly decreased (to 10%) when examining the results of conventional microbiological diagnostic on samples collected from the same patients within 7 days prior to or after the index sample [60]. Detection of bacteria by culture in subsequent samples collected from symptomatic patients who had initial PCR-positive, culture-negative detections, suggests that the molecular assays may be able to identify pathogens earlier in the course of infection. Rabin et al. reported on the results of a prospective observational cohort study of mechanically ventilated patients with suspected pneumonia [61]. They found that, among culture-negative BAL samples, those which were PCR-positive as assessed by the FilmArray Pneumonia panel had significantly higher white blood cell count, neutrophil count and amylase levels, suggestive of pneumonia, than the PCR-negative ones. Analysis of patients who had initial PCR-positive culture-negative results, who also did not have prior culture-positive results for the PCR-detected pathogen, showed that 39% of them had, upon repeat BAL collection, positive culture findings for the bacteria initially detected by molecular tests. Consistent with the idea that fastidious microorganisms are better detected by molecular than conventional diagnostics, Enne et al. reported that 86% of the pathogens detected by PCR but not by routine culture, were actually recovered by specialized culture methodology [54].

Although not reported by all studies, the proportion of patients who receive antibiotics prior to hospital admission may vary, from approximately 40–50% [62,63] up to around 80% [64,65]. In the case of antibiotic treatment onset prior to LRT sample collection, potential pathogens in the sample may lose viability in culture [66], whereas their DNA is still detected by the molecular method. This phenomenon inflates the apparent false clinical positive results (although true analytical positives) and decreases the calculated specificity of the molecular assays. The extent to which this situation occurs varies among studies [67], but may be responsible of up to half of the so-called false positives, wherein a bacterial pathogen is detected by PCR but not by conventional methods. For example, Buchan et al. observed that 49.3% of the culture-negative molecular detections in their study were made in patients who had received an antibiotic with potential activity against the specific bacterial target, within the 72 h preceding specimen collection [68]. In another study [45], the authors explained the additional bacterial findings of the PCR method over the culture-based diagnostic by the fact that most of the patients from the ICU who were enrolled in the study had received antibiotics on the day of the BAL procedure, which translated into low bacterial recovery in culture. The frequency of bacterial detection by routine culture was significantly higher in patients not receiving (81.8%) than in those receiving antibiotics (50.0%). The multiplex-PCR panel positive, culture-negative samples were reanalyzed for bacterial targets by in-house PCR, which confirmed the initial molecular detections. It was proposed that molecular diagnostic tests should be adapted to be able to differentiate living from dead bacteria [69]. Although inducing selective degradation of DNA in dead bacteria prior to sample DNA purification, leading to sole detection of living bacteria including VBNC cells, is feasible [70,71,72], we are not aware of the implementation of this approach in commercial molecular assays for respiratory samples.

Some of the bacteria detected by molecular assays may not be involved in the pathological process, but act as simple colonizers. One retrospective study conducted in Switzerland on 1078 BAL samples from 840 hospitalized patients (including 175 with pneumonia) reported low association between multiplex PCR detection of bacteria and the diagnosis of pneumonia at discharge [56]. In that setting, the odds ratio of pneumonia was only 1.1 (95% CI 0.7–1.6) for FilmArray Pneumonia Panel positive only, compared with 2.6 (95% CI 1.3–5.3) for culture-positive only, and 1.6 (95% CI 1.0–2.4) for PN-panel and culture-positive. *Haemophilus influenzae*, *Staphylococcus aureus*, and *Moraxella catarrhalis* were frequently detected by the panel, but were not associated with pneumonia. In contrast, panel detection of *Enterobacterales* and *P. aeruginosa* was associated with increased risk of pneumonia (odds ratios of 1.7 and 2.9, respectively).

The semiquantitative results offered by the FilmArray panels may be used by some laboratories to distinguish between colonizers and bacteria involved in the infective process. However, establishing a threshold to distinguish colonization from infection is challenging in practice, so clinical application of semiquantitative molecular results lacks consensus [66]. Variable agreement rates between molecular and culture quantitation in respiratory samples have been reported.

In a study on 212 samples from 150 patients suspected of bacterial pneumonia, in which 202 bacterial organisms were identified in BAL and ETA samples, Posteraro et al. reported complete quantitative agreement between the FilmArray Pneumonia Plus panel and culture in 35.1% of cases [73]. In 56% of the cases, bacterial loads reported by the FilmArray were higher than culture by at least 1 log_10_, whereas 8.9% were lower. In their comprehensive study, Ginocchio et al. found similar rates of concordance, with 25% of semiquantitative results obtained by the FilmArray being equal to those from culture, 70% higher than the culture, and 5% lower. Molecular results obtained on sputum or ETA samples were, on average,1 log_10_ higher than the corresponding culture results, whereas those from BAL samples were, on average, 1.5 log_10_ higher [51]. Ferrer et al. showed that concordance between molecular and conventional detection of pathogens increased with bacterial load [74]. In samples quantified as 10^7^ genomic equivalents/ ml by FilmArray, the PPA with culture was 88%, whereas in samples with 10^4^ genomic equivalents/ ml, PPA with culture was 45%. A result of 10^5^ genomic equivalents/mL by FilmArray predicted a positive culture with a sensitivity and specificity of 89% and 79.5%, respectively. Conversely, bacterial species recovered in clinically significant counts (≥10^5^ CFU/mL) usually yielded 10^7^ genomic copies/mL in FilmArray. Concordance between molecular and conventional methods was higher in the case of Enterobacterales than in non-fermenting GNB and in Gram-positives [74].

## 6. Clinical Utility of Molecular Testing in GNB Pneumonia

Molecular tests are considered as “the most accurate pathogen-based diagnostic tests currently available” [10], but their format and implementation influence their accuracy. For example, a systematic analysis found that the sensitivity of single-plex PCR assays was higher than that of multiplex PCR, with similar specificity [10]. Similarly, the overall sensitivity of in-house assays seems to be higher than that of commercial tests [75]. However, unlike rapid multiplex panels designed for point-of-care testing, performance of laboratory-developed tests or the high number of individual reactions required by single-plex testing are confined to central laboratories, with experienced specialists. In these settings, samples are usually processed in relatively large batches, and results may be available within 1–2 days, depending on service availability at weekends, or in a 24-h format.

Rapid molecular assays should be sensitive and specific, but diagnostic accuracy does not automatically translate into clinical advantage. Given that with each hour of delayed appropriate antibiotic treatment the clinical prognosis of the patients with severe infections becomes poorer [76], even tests which are not as accurate as the gold standard may provide clinical utility if their results are available fast enough [77]. It may be the case that, in rapidly evolving infections, early information is more beneficial for the patient than complete, but delayed, information [69]. Rapid reporting might add significantly to the quality of anti-infective treatment. Shortening the time for antibiotic initiation and downgrading to a drug with a narrower spectrum of activity are desiderata which should be pursued [78]. Thus, the greatest clinical benefit of molecular multiplex panel testing may be to enable a better use of antibiotics [79].

The recent COVID-19 pandemic brought an increase in antibiotic usage, paralleled by an apparent rise in antimicrobial resistance [80,81]. It was estimated that antibiotics were administered to nearly 80% of the patients with COVID-19 admitted to the ICU [20]. In the meanwhile, bacterial co-infections were documented in a minority of cases. One meta-analysis reported that, among hospitalized COVID-19 patients, laboratory-confirmed bacterial co-infections were found in 7% (95% CI 3–12%), whereas subgroup analysis on ICU-admitted patients found evidence of bacterial co-infections in 14% (95% CI 5–26%) [82]. Besides the public health concerns, needless use of antibiotics causes adverse effects and additional expense [69].

In some bacteria, identification of resistance mechanisms enables targeted use of novel beta-lactam/ beta-lactamase inhibitors (BL/BLIs) with differential activity against carbapenem-resistant GNB producing different types of carbapenemases (e.g., cetazidime-avibactam, active against Ambler class A and D carbapenemases, KPC and OXA-48, respectively), with clinical benefit [83].

The high sensitivity of molecular panels, coupled with the high negative predictive values for the detection of bacteria, enables these tests to exclude the bacterial etiology of pneumonia with high accuracy. This may prove advantageous, since viral infections are frequent in CAP. For example, in 384 patients with hospitalized CAP, Markussen et al. found that only 32.8% had solely bacterial infections, but almost 48.5% had one or more respiratory viruses detected [84]. In another study, among 2259 patients with radiographic evidence of pneumonia, viral infections were detected in 23% [2].

Negative results for typical bacteria, concomitant with the detection of viruses and atypical bacteria as etiological factors in pneumonia patients, corroborated by low levels of serum biomarkers such as procalcitonin [85], especially in patients with CAP, may limit the use of antibiotics [79]. In a retrospective study on LRT samples from 323 hospitalized patients with clinical and radiological evidence of CAP, Gadsby et al. estimated that, based on molecular testing, de-escalation of initial empiric antibiotic treatment could have been achieved in 77.2% of cases. Most situations of possible de-escalation consisted of switching from an amoxicillin–clavulanate combination to amoxicillin when *S. pneumoniae* or *H. influenzae* were detected, and withdrawal of clarithromycin in cases where atypical bacteria were not detected by PCR [86]. A randomized controlled trial (RCT) with 800 patients diagnosed with LRTI investigated whether the addition of a rapid molecular assay for respiratory pathogens could improve clinical management of these patients. The results of the molecular panel (FilmArray) were communicated and explained to clinicians, once available. In patients with rapid molecular testing, the duration of intravenous antibiotics, length of hospital stay, and cost of treatment were reduced compared with the group of patients with a routine laboratory diagnostic [87].

### 6.1. Clinical Utility of Rapid Multiplex PCR Assessed in Randomized Controlled Trials

Poole et al. reported on the results of a single-center, parallel group, open-label RCT conducted in the UK to assess the impact of syndromic molecular point-of-care testing compared to conventional diagnostic testing, on antibiotic use [88]. Two hundred patients with LRTI (CAP, HAP or VAP) from the critical care unit were randomly assigned to a point-of-care testing (using FilmArray Pneumonia Plus panel) or to a control group. In the case of patients in the intervention group, LRT samples were immediately analyzed with the molecular panel, and clinical advice including antibiotic prescribing guidance, taking into account patients’ microbiological history and clinical status, was offered by clinical infection specialists to the treating clinicians, as soon as the results of FilmArray testing became available. Recommendations on antibiotic treatment were not mandatory and pertained to the treating clinician’s decision. In patients from the control arm, microbiological investigation including on-request multiplex respiratory virus testing, representing standard clinical care, was at the discretion of the responsible clinical team. The FilmArray identified typical bacterial pathogens with 93.3% PPA and 95.4% NPA and results were available with a median of 65 h earlier, compared with standard diagnostics. Time to results-directed therapy, as well as time to de-escalation was more than 40 h shorter in the mPOCT group than in the control group. In the mPOCT group, the probability of receiving results-guided therapy was two times higher, and de-escalations were five-times more frequent than in the control group (Table 4). Most de-escalations enabled by mPOCT consisted of the narrowing of antibiotic spectrum of the beta-lactam therapy (changing from piperacillin–tazobactam to amoxicillin–clavulanate) in the absence of *P. aeruginosa*, and withdrawal of macrolides when no atypical bacteria were detected. Escalations to broader-spectrum beta-lactams (from amoxicillin–clavulanate to piperacillin–tazobactam or carbapenems) upon pathogen detection, were possible on average 1 day earlier in patients receiving mPOCT testing than with conventional diagnostics. Moreover, the use of mPOCT as guidance appeared to be safe, with no changes in mortality.

Another single-center, parallel group, single blinded RCT, conducted in Norway with 374 patients suspected of CAP, showed that molecular testing (FilmArray Pneumonia Plus panel) significantly increased the proportion of patients who received pathogen-directed treatment. The median time to pathogen-directed treatment was reduced by 9.4 h, compared with standard of care [89]. There too, the time to results availability was much shorter (by about 54 h) in the mPOCT group than in standard-of-care group. Pathogen-directed adjustment of treatment within the first 48 h was approximately three times more likely in the mPOCT group than in the controls. The continuation of appropriate empirical treatment, as well as cessation of antibiotics after the first dose, were more frequent in the mPOCT group (Table 4).

The rapid use of mPOCT panels is facilitated by the early collection of LRT specimens. Serigstad et al. showed that induced sputum or endotracheal aspirates could be successfully collected in the Emergency Department, leading to large improvement in the time-to-results, both for microbiological and for molecular testing [90].

In a multicenter RCT conducted at two tertiary care centers in Switzerland, the duration of inappropriate antibiotic treatment in pneumonia patients was reduced by 38.6 h (equivalent to a 45% reduction) by multiplex PCR testing (Unyvero Hospitalized Pneumonia Cartridge), compared with conventional microbiology [91]. In the PCR group, an indication to change the initial antibiotic treatment according to the multiplex panel results was made in the case of 66% of patients. Seventy-five percent of those recommendations were followed by the attending clinician.

Certain differences between the recommendation of the clinical infection specialist based on the rapid testing results and the decision of the treating clinician were also observed elsewhere [92]. For example, in the case of severely ill patients, neither the negativity of the multiplex panel (FilmArray Pneumonia Panel) or that of the bacterial culture led to de-escalation or withdrawal of antibiotics in approximately half of the cases of ICU-admitted COVID-19 patients, in one study [64]. However, the authors concluded that the FilmArray panel was useful in the antibiotic stewardship, since a negative test resulted in no prescription or withdrawal of antibiotics in 28% of patients with suspicion of pneumonia. The authors of another study were unable to demonstrate a reduction in overall antibiotic exposure based on the combination of multiplex PCR and procalcitonin results in 194 critically ill COVID-19 patients [93]. Although the number of antibiotic-free days within one week after randomization was higher in the intervention than in the control arm (4 days vs. 2 days), antibiotics were reintroduced and, at day 28 from randomization, the difference in antibiotic-free time between the two groups was no longer significant.

A parallel-group, open-label, multicenter RCT was conducted on 294 patients with CAP from three Danish centers [94]. The utility of the results from Multiplex PCR (FilmArray Pneumonia Plus Panel) was compared with that of SOC. Regarding prescriptions for no or narrow-spectrum antibiotics at 4 h after admission, no differences were found. However, prescriptions in the mPCR group were more targeted at 4 h and 48 h, and more appropriate at 48 h and on day 5 from admission, compared with SOC. No differences in mortality, ICU admission or hospital readmission within 30 days were observed, inferring the safety of the mPCR-directed antibiotic therapy.

### 6.2. Contributors to Clinical Utility of Molecular Syndromic Panels in Pneumonia

Clinical utility of molecular testing may be optimized by adoption of certain implementation strategies, such as pre-analytical restrictions, measures to shorten turn-around-times, and better interpretation of results in a clinical context.

Pre-analytical factors include test selection appropriateness, patient characteristics, sample type, collection technique, and storage.

The main difference between the FilmArray Pneumonia (plus) and the Unyvero Hospitalized Pneumonia panels is that the former is able to detect bacteria and viruses, whereas the latter detects *S. maltophilia*, but no viruses. Diagnosis of viral infection enables de-escalation or withdrawal of antibiotic therapy, whereas detection of *S. maltophilia* allows for targeted therapy with co-trimoxazole [54]. Early diagnostic of viral infection and exclusion of bacteria from the etiology of pneumonia in children may also lower the need for chest radiography, as shown by a Cochrane review [95].

As a factor of pre-test probability, the population to which a patient belongs should be considered [5]. The etiology of pneumonia varies widely with clinical scenario (CAP, HAP, or VAP), geographic region, setting and time. While molecular panels typically cover pathogens responsible for pneumonia, they may not be equally well suited for all situations. For example, patients with chronic obstructive pulmonary disease or cystic fibrosis may be colonized by bacteria which are more frequently associated with HAP. In consequence, these agents may cause community-acquired infections in such patients [79]. Adherence to clearly defined clinical criteria for pneumonia [96] when selecting patients to be tested by multiplex PCR improves assay specificity and meaningfulness of results. Also, the use of PCR platforms is considered justified in the case of immunocompromised patients, for whom specific therapeutic options are available, or in the presence of severe clinical manifestations [97].

Ideally, tests should provide information relevant to the choice of treatment and their results should be available within 6 h from specimen collection [69]. With their analytical time of under 5 h, multiplex PCR platforms are able to deliver these fast turn-around-times and provide results with improved speed compared with standard microbiological diagnostics, as shown in clinical studies [88,89]. However, practical issues such as batch analysis of samples or staff availability outside the usual working hours, may introduce considerable delays under real-world conditions [69].

The question of the most appropriate sample type for the etiologic diagnostic of pneumonia is still debated. For the diagnostic of VAP, the American Thoracic Society/Infectious Diseases Society of America (ATS/IDSA) guidelines suggest the use of endotracheal aspirate samples [98]. In contrast, ERS/ESICM/ESCMID/ALAT recommend invasive diagnosis with bronchoscopy [99]. The 2023 ERS/ESICM/ESCMID/ALAT Guidelines for the Management of Severe community-acquired pneumonia suggest that an LRT sample (sputum or endotracheal aspirate) should be collected for multiplex PCR if this kind of testing is available, whenever non-standard antibiotics for this condition are prescribed or considered [100]. It is worth noting that sputum samples may be contaminated with upper respiratory-tract flora. Tracheal aspiration for obtaining respiratory tract samples is less invasive, less expensive, easier to perform, and does not require specialized staff, compared with BAL, which is obtained through fiberoptic bronchoscopy, a more invasive procedure requiring highly trained personnel and more time to perform [15]. The less-invasive mini-BAL, which is performed blindly, without a bronchoscope, was shown to have high diagnostic concordance with BAL in HAP patients [101]. Also, concordance between qPCR and conventional culture on ETA in suspected VAP patients minimizes the utility of BAL [102].

LRT samples such as ETA and BAL fluid are less vulnerable to contamination than sputum, but they are not part of routine use in non-intubated patients. Sputum, on the other hand, was shown to offer higher diagnostic accuracy for molecular detection of both MSSA and MRSA [75]. It is also more frequently contaminated by oropharyngeal flora, leading to numerous false positive results in multiplex PCR, as pointed out by a large meta-analysis [53]. Although sputum collection is known to be relatively difficult to obtain, with success rates around 30–60%, Serigstad et al. managed to increase the rate of successful sample collection by induced sputum to 95% in the Emergency Department, with around 40% representative samples by microscopy, showing that the procedure is feasible and well tolerated [90]. Both sputum and BAL fluid samples are viscous, a characteristic that may render nucleic acid purification difficult. Some platforms, such as the QiaStat-DX, allow the dilution of such samples in order to increase their fluidity [103,104], whereas others, such as the FilmArray do not [41]. However, nucleic acid extraction procedures have become efficient enough, either in commercial or in laboratory-developed format [105], to allow successful detection by PCR.

Prolonged sample storage for several months decreases detection sensitivity for respiratory pathogens, as seen by Gadsby et al. [45], whereas storage for up to 72 h of either fresh or frozen samples does not significantly impact analytical sensitivity of PCR [54]. However, with the exception of retrospective studies, sample storage for several days negates the utility of rapid molecular testing in the clinical setting, as the speed advantage over the classical diagnostic methods is lost.

Counseling by clinical infection specialists regarding the application of results provided by multiplex panels seems to improve antibiotic stewardship practices. Two RCTs enrolling approximately the same number of patients investigated the use of a molecular panel (FilmArray Respiratory Panel) in LRT infections. One of them [87], wherein PCR testing results were explained to clinicians by microbiologists, showed a higher number of de-escalations and a shorter duration of antibiotic treatment associated with the use of the FilmArray. In the second trial [106], no reduction in the antibiotic therapy duration or in the proportion of patients receiving antibiotics was observed, when the molecular diagnostic device was placed at the patient point-of-care and the results were directly retrieved by the treating physicians. Molecular testing is already perceived as becoming too complex for non-infectious-disease physicians [107,108].

A potential risk in the use of more sensitive PCR testing in pneumonia is that the higher number of microorganisms detected in comparison with routine diagnostic would lead to an unnecessary increase in antibiotic prescriptions, as suggested by the results of Prinzi et al. In their retrospective study of medical records of 448 mechanically ventilated pediatric patients, the authors observed that microorganism over-reporting in ETA cultures was associated with a more than two-fold increase in antibiotic prescriptions, compared with cases in which guideline-concordant reporting of microbiological findings was performed [109]. A more nuanced, even selective, reporting of results was thus proposed [66]. Another approach involved the use of automatic alerts issued within the electronic health records of patients with serum procalcitonin <0.25 ng/mL, virus detected on respiratory PCR, and active use of systemic antibiotics. The automatic message prompted the clinicians that the laboratory results suggested a viral infection and suggested that they reassess the necessity for antibiotics. In this multisite study, conducted at five hospitals, a reduction in antibiotic treatment by more than 2 days was achieved by the alerting system [110]. Finally, and not least, the probable significance of the detected microorganisms (stratified as “proven”, “probable”, and “uncertain” causes of pneumonia), following earlier published models [84,96,111], as well as the suggestion of targeted or empiric treatment regimens [3,112], should be provided to the attending clinicians.

## 7. Concluding Remarks

Novel molecular rapid assays for the identification of respiratory pathogens and prediction of their antimicrobial susceptibility phenotypes have the potential to revolutionize the diagnostics and treatment of pneumonia, by enabling physicians to institute early targeted therapy and to reduce the misuse of antibiotics. They show good concordance with conventional microbiology regarding the identification of pathogens and prediction of their antibiotic resistance profiles, but provide results faster and with higher analytical sensitivity. Unlike conventional microbiology, molecular methods can identify fastidious or unculturable pathogens, including VBNC cells, and are less affected by prior antibiotic treatment in patients.

So, will multiplex molecular assays replace the conventional microbiological investigation in the diagnostic of GNB pneumonia?

No… we believe that, at least in the near future, pneumonia diagnostic cannot rely solely on molecular methods. Assays based on targeted amplification can only detect pathogens and resistance markers that were included in their design, meaning that less-frequently involved microorganisms will likely be missed out. Also, the unwanted effect of their high sensitivity is that molecular assays may frequently detect agents which are not involved in the infectious process, but are simple colonizers. To be fair, the distinction between colonizers and pathogens is not entirely solved by culture-based microbiology, either. As for the pathogen susceptibility to antimicrobials, there are far more resistance mechanisms than can be detected by the targeted genetic approach available in present multiplex assays. Another issue remaining to be solved is the accurate attribution of the detected resistance markers to specific detected pathogens in the sample. For now, detection of both pathogens and resistance genes is pooled by sample, meaning that even if a resistance marker is detected, we cannot know precisely to which bacteria it belongs.

But… molecular assays may make a good addition to the existing standard of care in pneumonia diagnostic. Clinical trials show the potential of this technology to improve antibiotic stewardship. Patients whose samples are tested by molecular assays are more likely to receive targeted therapy within the first few hours after sample collection, with shorter times to targeted therapy and earlier results-directed therapy change (escalation or de-escalation). Clinical utility of molecular methods can be further enhanced by knowledge of local epidemiology of resistant strains, availability of testing equipment at the point of care or the emergency department, adequate counseling of the attending clinician by the clinical infection or microbiology specialist regarding result interpretation, and choice of antibiotic treatment.

Still information on practical issues such as the possible prioritization of specific patient categories at risk, in order to maximize cost effectiveness, and the proper way of formulating results and guidance on their interpretation, remain to be explored in future studies.

## Figures and Tables

**Table 1 antibiotics-13-00805-t001:** Molecular targets of commercial multiplex PCR pneumonia panels.

Target Category	Target	BioFire FilmArray Pneumonia (Plus) Panel	Unyvero Hospitalized Pneumonia Panel
Bacteria			
*Enterobacteriaceae*	*Citrobacter freundii*	-	Y
	*Enterobacter cloacae* complex	Y	Y
	*Escherichia coli*	Y	Y
	*Klebsiella aerogenes* (*Enterobacter aerogenes*)	Y	Y
	*Klebsiella oxytoca*	Y	Y
	*Klebsiella pneumoniae*	*K. pneumoniae* group (*K. pneumoniae* (KPI), *K. quasipneumoniae* (KPII), *K. variicola* (KPIII))	Y (*K. pneumoniae* (KPI), *K. quasipneumoniae* (KPII))
	*Klebsiella variicola*	-	Y
	*Morganella morganii*	-	Y
	*Proteus* spp.	Y	Y
	*Serratia marcescens*	Y	Y
Non-fermenting Gram-negative bacteria	*Acinetobacter baumannii* complex	Y	Y
	*Moraxella catarrhalis*	Y	Y
	*Pseudomonas aeruginosa*	Y	Y
	*Stenotrophomonas maltophilia*	-	Y
Atypical bacteria	*Chlamydia pneumoniae*	Y	Y
	*Legionella pneumophila*	Y	Y
	*Mycoplasma pneumoniae*	Y	Y
Gram-positives	*Staphylococcus aureus*	Y	Y
	*Streptococcus agalactiae*	Y	-
	*Streptococcus pneumoniae*	Y	Y
	*Streptococcus pyogenes*	Y	-
Fungi	*Pneumocystis jirovecii*	-	Y
Viruses	Adenovirus	Y	-
	Coronavirus	Y	-
	Human metapneumovirus	Y	-
	Human rhinovirus/enterovirus	Y	-
	Influenza A virus	Y	-
	Influenza B virus	Y	-
	Middle East respiratory syndrome coronavirus (MERS-CoV)	Y *	-
	Parainfluenza virus	Y	-
	Respiratory syncytial virus	Y	-
Antimicrobial resistance genes	Carbapenemase, class A	*bla* _KPC_	*bla* _KPC_
	Carbapenemase, class B	*bla* _IMP_	*bla* _IMP_
		*bla* _NDM_	*bla* _NDM_
		*bla* _VIM_	*bla* _VIM_
	Carbapenemase, class D	-	*bla* _OXA-23_
		-	*bla* _OXA-24_
		*bla*_OXA-48_–like (*oxa*-48, -162, -181, -199, -204, -232, -244, -245, -252, -370, -484, -505)	*bla*_OXA-48_ (*oxa*-48, -162, -181, -232, -244)
		-	*bla* _OXA-58_
	Extended spectrum β-lactamase	*bla* _CTX-M_	*bla* _CTX-M_
	Fluoroquinolone resistance	-	*gyr*A83 of *E. coli* and *P. aeruginosa*
		-	*gyr*A87 of *E. coli* and *P. aeruginosa*
	Macrolide/ lincosamide resistance	-	*erm*B
	Methicillin resistance	*mec*A/C and MREJ	*mec*A, *mec*C
	Penicillinase	-	*bla* _tem_
		-	*bla* _shv_
	Sulfonamide resistance	-	*sul*1

* Only in BioFire FilmArray Pneumonia Plus panel.

**Table 2 antibiotics-13-00805-t002:** Descriptive characteristics of 12 studies assessing the performance of multiplex PCR assays relevant to the diagnostic of pneumonia caused by Gram-negative bacteria.

Publication	Assay	Time	Location	Study Type	Population	No. of Patients	Antibiotic before Sampling	No. of Samples	Sample Type
Gadsby et al., 2019	Unyvero P55 Pneumonia	2013–2015	UK	retrospective	ICU patients	74	63.80%	74	BAL
Klein et al., 2021	Unyvero Lower Respiratory Tract BAL	2015–2018	USA	multicenter, retrospective + prospective			NA	1016 prospective; 392 archived	BAL
Collins et a.l, 2020	Unyvero Lower Respiratory Tract		USA			98	NA	175	BAL
Enne et al., 2022	Unyvero Hospitalized Pneumonia	2016–2018	UK	multicenter, prospective	nosocomial pneumonia (HAP/VAP)	260 (suspected HAP); 392 (suspected VAP)	43.40%	652	Sputum (272) ETA (n = 299) BAL (44) mini-BAL (23)
Peiffer-Smadja	Unyvero Hospitalized Pneumonia	2017–2018	France	single-center, prospective	ventilated HAP or VAP	85 patients, 95 episodes (24 ventilated HAP or 71 VAP)	43.20%	95	BAL (72) mini-BAL (23)
Luyt et al., 2020	Unyvero P55 Pneumonia, Hospitalized Pneumonia	2016–2019	France	prospective	suspicion of VAP	83	81.00%	93	BAL
Buchan et al., 2020	BioFire FilmArray Pneumonia	2016–2017	USA	multicenter, retrospective	suspicion of HAP or VAP		NA	259	BAL (237) mini-BAL (22)
Enne et al., 2022	BioFire FilmArray Pneumonia	2016–2018	UK	multicenter, prospective	nosocomial pneumonia (HAP/VAP)	260 (suspected HAP); 392 (suspected VAP)	42.50%	652	Sputum (272) ETA (299) BAL (44) mini-BAL (23)
Gastli et al., 2021	BioFire FilmArray Pneumonia	2018	France	multicenter, prospective	suspicion of pneumonia	515	NA	515	Sputum (58) ETA (217) BAL (240)
Posteraro et al., 2021	BioFire FilmArray Pneumonia plus	2020–2021	Italy	single-center, prospective	COVID-19 patients with suspicion of VAP	150	NA	212	ETA (130) BAL (82)
Ginocchio, 2021	BioFire FilmArray Pneumonia plus		Europe (12 countries), Israel	multicenter	suspicion of pneumonia		NA	2476	Sputum-like (1242) BAL-like (1234)
Kamel et al., 2022	BioFire FilmArray Pneumonia plus	2021	Egypt	single-center, prospective	suspicion of HAP or VAP		NA	50	Mini-BAL
Gong et al., 2024	BioFire FilmArray Pneumonia	2021–2023	China	single-center, prospective	patients diagnosed with acute LRTIs (CAP and HAP)	130 (CAP); 57 (HAP)	NA	187	BAL

Abbreviations: BAL = bronchoalveolar lavage; ETA = endotracheal aspirate; HAP = hospital-acquired pneumonia; ICU = intensive care unit; VAP = ventilator-associated pneumonia; NA = data not available

**Table 3 antibiotics-13-00805-t003:** Diagnostic performance of multiplex PCR assays for the identification of bacterial etiology of pneumonia and prediction of antimicrobial resistance of identified pathogens.

Publication	Assay	Pathogen Identification				Identification of Antimicrobial Resistance		
		Reference Test	PPA	NPA	Out-of-Panel Microorganisms/Tested Samples	Reference Methodology for Phenotypic Testing	PPA	NPA
Gadsby et al., 2019	Unyvero P55 Pneumonia	MALDI-TOF and in-house bacterial PCR	56.9%	58.5%		Vitek 2 (UK standards)	18.8%	94.9%
Klein et al., 2021	Unyvero Lower Respiratory Tract BAL	MALDI-TOF	93.4%	98.3%	28/1016	Vitek 2, Phoenix, MicroScan, Sensititre, disk diffusion, broth dilution, or agar dilution (CLSI)	insufficient data for PPA and NPA	
Collins et al., 2020	Unyvero Lower Respiratory Tract	MALDI-TOF	96.5%	99.6%	35/175	disk diffusion (CLSI)	insufficient data for PPA and NPA	
Enne et al., 2022	Unyvero Hospitalized Pneumonia	UK standard	90.7%	96.8%	16/652			
Peiffer-Smadja	Unyvero Hospitalized Pneumonia	MALDI-TOF	80%	99%	8/95	disk diffusion (EUCAST)	77%	99%
Luyt et al., 2020	Unyvero P55 Pneumonia, Hospitalized Pneumonia	culture	77.4%	14.3%	5/93	disk diffusion (French standard)	46.3%	82.7%
Buchan et al., 2020	BioFire FilmArray Pneumonia	MALDI-TOF	96.2%	98.1%	30/259	Vitek 2, BD Phoenix, disk diffusion, E-test	52.4%	66.7%
Enne et al., 2022	BioFire FilmArray Pneumonia	UK standard	96.7%	95.8%	28/652			
Gastli et al., 2021	BioFire FilmArray Pneumonia	Vitek2 and/or MALDI-TOF	94.4%	96.0%	46/515	Vitek2 or disk diffusion (EUCAST)	92.3%	99.5%
Posteraro et al., 2021	BioFire FilmArray Pneumonia plus	MALDI-TOF	100%	99.2%	2/120	Vitek 2, broth microdilution	100%	99.7%
Ginocchio, 2021	BioFire FilmArray Pneumonia plus	standard of care (varied with site)	92.9%	96.10%	512/2476	standard of care (varied with site)	insufficient data for PPA and NPA	
Kamel et al., 2022	BioFire FilmArray Pneumonia plus	Vitek 2	100%	90%	2/50	Vitek 2	97%	95%
Gong et al., 2024	BioFire FilmArray Pneumonia	Vitek 2	85%	92%	4/187	broth dilution (CLSI)	85%	60%

Abbreviations: MALDI-TOF = matrix-assisted laser desorption/ionization–time-of-flight mass spectrometry; NPA = negative percentage agreement; PPA = positive percentage agreement

**Table 4 antibiotics-13-00805-t004:** Randomized clinical trials investigating the utility of multiplex PCR panels for antibiotic stewardship in pneumonia caused by Gram-negative bacteria.

Publication	Assay	Time	Location	Study Type	Population	No. of Patients	Sample Type	Main Results
Darie et al., 2022	Unyvero Hospitalized Pneumonia	2017–2019	Switzerland	Multicenter RCT	Suspicion of pneumonia (CAP, HAP) and risk of GNB infection	208: - I: HPN + antibiotic treatment guidance + SOC (n = 100) - C: SOC (n = 108)	BAL	*Primary outcome:* time (h) on inappropriate antibiotic therapy from bronchoscopy to day 30 or to discharge or: 38.6 (I) vs. 85.7 (C) * *Secondary outcomes:* - time (days) to clinical stability or discharge: 2.5 (I) vs. 2.4 (C), NS - proportion of patients reaching clinical stability or being discharged: 90% (I) vs. 92% (C), NS - performance of mPCR to detect GNB: sensitivity 55.6%, specificity 86.6%.
Poole et al., 2022	FilmArray Pneumonia Plus	2019–2021	UK	Single-center, parallel-group, open-label RCT	CAP, HAP, VAP	200 patients (100 per arm): - intervention (I): FAPP + antibiotic prescription guidance + SOC - control (C): SOC	ETA: 63% Sputum: 29% Directed BAL: 4% Undirected BAL: 4%.	*Primary outcome:* proportion of patients receiving results-directed therapy within 48 h from result: 80% (I) vs. 29% (C) * *Secondary outcomes:* - time to results: 1.7 h (I) vs. 66.7 h (C) * - time to result-directed therapy: 2.3 h (I) vs. 46.1 h (C) * - results-directed antibiotic de-escalation: 42% (I) vs. 8% (C) * - time to de-escalation: 4.8 h (I) vs. 46.5 h (C) * - results-directed antibiotic escalation: 9% (I) vs. 1% (C) * - credible pathogen-identification rate: 71% (I) vs. 51% (C) * - similar safety outcomes in I vs. C
Fartoukh et al., 2023	FilmArray Pneumonia/ FilmArray Pneumonia Plus	2020	France	Multicenter, parallel-group, open-label RCT	Critically-ill ICU patients with SARS-CoV-2 pneumonia	191: - I: PCT + mPCR algorithm + routine (n = 93) - C: PCT + routine (n = 98)	Sputum ETA BAL	*Primary outcome:* number of days free of antibiotic by day 28 after randomization: 12 (I) vs. 14 (C), NS *Secondary outcomes:* - number of days of antibiotic exposure: 9 (I) vs. 8 (C), NS - cumulative antibiotic duration (days): 11 (I) vs. 10 (C), NS
Cartuliares et al., 2023	FilmArray Pneumonia Plus	2021–2022	Denmark	Multicenter, parallel-group, open-label RCT	Patients admitted with suspicion of CAP	294: - I: FAPP + antibiotic treatment guidance + SOC (n = 148) - C: SOC (n = 146)	Sputum ETA	*Primary outcome:* proportion of patients with prescription of “no or narrow-spectrum” antibiotics within 4 h after admission: 62.8% (I) vs. 59.6% (C), NS *Secondary outcomes:* - targeted antibiotic therapy at 4 h from admission: 57.7% (I) vs. 24.1% (C) * - adequate antibiotic at 48 h from admission: 76.9% (I) vs. 62.1% (C) * - similar rates of ICU admission, readmission within 30 days, in-hospital or 30-day mortality, in I and C
Markussen et al., 2024 (*JAMA*)	FilmArray Pneumonia Plus	2020–2022	Norway	Single-center, parallel-group, single-blinded RCT	CAP (Emergency Department)	374 (187 per arm): - I: FAPP + SOC - C: SOC	Sputum induction, ETA	*Primary outcomes:* - proportion of patients receiving results-directed treatment within 48 h from randomization: 35.3 (I) vs. 13.4% (C) * - time to provision of pathogen-directed treatment within 48 h from randomization: 34.5 h (I) vs. 43.8 h (C) * *Secondary outcomes:* - TAT was 53.8 h shorter in I than in C - results-directed antibiotic de-escalation: 10.3% (I) vs. 4.9% (C) * - results-directed antibiotic escalation: 14.4% (I) vs. 3.9% (C) * - pathogen detections: 175 bacteria and 74 viruses in I vs. 72 bacteria and 63 viruses in C * - similar safety outcomes in I vs. C

Abbreviations: * = significant difference; C = control; FAPP = FilmArray Pneumonia Plus; GNB = Gram-negative bacteria; I = intervention; mPCR = multiplex PCR; NS = not significant difference; PCT = procalcitonin; SOC = standard of care.

## Data Availability

Data sharing is not applicable.

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
