# Peer review of "Rapid Molecular Diagnostics of Pneumonia Caused by Gram-Negative Bacteria: A Clinician’s Review"

_antibiotics, 2024, doi:10.3390/antibiotics13090805_

Round 1

Reviewer 1 Report

Comments and Suggestions for Authors

Dear Authors,

Your Review paper entitled "Rapid molecular diagnostic of pneumonia caused by Gram-negative bacteria. A clinician’s review" has been carefully revised.

This Review deserves attention since it highlights an important topic from medical point of view, this paper talked about the importance of using molecular diagnostic tools instead/or with the routine test "Bacterial Culture" following by "Disk Diffusion Method for Antibiogram" for the identification of the causative bacterium and its antimicrobial resistance profile. In short time and In low cost comparing to routine technics.

This paper is excellent, it is well written, well designed, and well presented, the English language is also excellent.

Kindly find below the list of my comments, remarks and Suggestions. (Minors and Majors)

Minors:

01- In the affiliation section, I suggest to remove the email addresses of all authors, and leave the corresponding author's email address.

02- In the Keywords section, I suggest to add Antimicrobial Resistance genes.

03- In the line 308, when authors used the term "superbugs", I prefer to replace it by "Multidrug Resistant Strains". 

04- When Authors talked about Klebsiella pneumoniae as cause of pneumoniae authors are invited to add the following article as reference for this point:

-- https://www.mdpi.com/2079-7737/13/2/78

05- When Authors talked about Acinetobacter baumannii as cause of pneumoniae authors are invited to add the following article as reference for this point:

-- https://pubmed.ncbi.nlm.nih.gov/18092011/

06- Line 376, When you mentioned "severe acute respiratory syndrome coronavirus 2" Authors are invited to put (SARS-CoV-2) after this term.

07- When authors talked about the excessive use of antibiotics during COVID-19, authors are invited to use the following articles as references:

-- https://www.mdpi.com/2673-8112/4/7/64

-- https://pubmed.ncbi.nlm.nih.gov/34316138/

08- In the Table 3, second row, last column why there is no number for the NPA since their is a number for the PPA?

09- When authors talked about negative points of bacterial culture as diagnostic tool, they are invited to talk about Viable But Not Culturable (VBNC) strains, since these strains can be detected by molecular tools but not by routine culture methods, they can use these articles as references for this point:

-- https://www.frontiersin.org/journals/public-health/articles/10.3389/fpubh.2014.00103/full

-- https://academic.oup.com/femsre/article/34/4/415/538375

Majors

01- When authors talked about the importance of Molecular Diagnosis, I think it is good to mention that this technic can detect the presence of mutations, such it was the case during COVID-19 pandemic (S-mutant types).

02- I think it is good to mention the importance of the COVID-19 diagnosis in establishing the "one step RT-PCR" as a GOLD STANDARD for diagnosis of an infectious diseases, then talk about applying RT-PCR as routine diagnosis of different infections, including "Pneumonia caused by Gram-Negative bacteria".

03- When comparing different PCR panels, why this review is limited to "BioFire Film Array Pneumonia (plus) panel" and "Unyvero Hospitalized Pneumonia panel"? 

04- In the Title, I Suggest to Replace "Clinical-Review" by "A General Review" since the present paper is not at 100% a Clinical Review.

BR,

Reviewer 2 Report

Comments and Suggestions for Authors

In this review, the authors summarized molecular diagnostic for identifying the respiratory pathogens of Gram-negative bacteria in pneumonia and predicting the antimicrobial susceptibility phenotypes to provide the treatment of pneumonia. In addition, the main strengths and limitations of these molecular rapid assays were presented in the clinical utility.

The manuscript is presented well. I have one concern about the molecular diagnostic method. Except for the PCR assay, how about other molecular diagnostic methods for Gram-negative bacteria detection? Such as fluorescence imaging and PET radio-imaging for bacterial infection.

Reviewer 3 Report

Comments and Suggestions for Authors

Rapid molecular diagnostic of pneumonia caused by Gramnegative bacteria. A clinician’s review

Abstract:

It is incomplete, it lacks information about the methodology.

Keywords:

Replace the words that already appear in the title of the manuscript.

Introduction

In this topic, it would be interesting for you to highlight microbial agents that cause pneumonia.

On line 74 check the word "Definitions". It's out of context

What is the methodology used in this narrative review of the study?

The tables are not standardized.

Table 2. Descriptive characteristics of 12 studies assessing the performance of multiplex PCR assays relevant to the diagnostic of pneumonia caused by Gram-negative bactéria. This is not formatted as a table, but as a frame.

Tabela 2. What does "NA" mean?

Table 3. This is not formatted as a table, but as a frame.

Table 4. This is not formatted as a table, but as a frame.

Concluding remarks

It needs to be improved or rewritten, as it does not represent the significance of the study;

References are not used in conclusions.

What is the main relevance and impact of this study? Score.

Reviewer 4 Report

Comments and Suggestions for Authors

The review article submitted for evaluation is very extensive and comprehensive. First, the authors present the classification and epidemiology of pneumonia, followed by a description of the mechanisms of pathogenicity and drug resistance of the most important Gram-negative bacilli. This is followed by an overview of commercial molecular tests used in the diagnosis of pneumonia. Finally, a narrative literature review is presented, including publications related to the diagnostic performance of molecular tests and their clinical utility. The construction of the article is logical, the literature cited is mostly from recent years. In my opinion the paper is of good quality and I support its publication.

Round 2

Reviewer 1 Report

Comments and Suggestions for Authors

Dear Authors,

Your revised form of the manuscript was carefully reviewed,

This paper is more suitable for publication in its present form,

Thanks to the modifications you made,

BR.

Reviewer 2 Report

Comments and Suggestions for Authors

The authors have done all the necessary correction and now the manuscript could be accepted in this current version.

Reviewer 3 Report

Comments and Suggestions for Authors

The manuscript has been improved